# OpenReview forum: "Measuring Value Understanding in Language Models through Discriminator-Critique Gap"
_ICLR.cc/2024/Conference — ICLR 2024 Conference Withdrawn Submission_

### Official Review · Reviewer_c54v · 2023-11-01

**Soundness:** 2 fair
**Presentation:** 1 poor
**Contribution:** 2 fair
**Rating:** 5
**Confidence:** 3

**Summary:**

The authors propose the Value Understanding Measurement (VUM) dataset (or framework), which assesses the "know what" and "know why" aspects of value comprehension in LLMs by measuring the gap between their responses and GPT-4 values. They use the Schwartz Value Survey to come up with diverse questions. They compare the target model’s responses to those questions with GPT-4 responses, and also use GPT-4 to compare the similarity between them. They examine five LLMs and finds that models perform well on the “know why” aspect and models gets better on the “know what” aspect as its size gets larger.

**Strengths:**

Probing LLMs for the understanding of values and their reasons can be interesting.

**Weaknesses:**

- The experiment results lack meaningful insights or patterns. While the radar charts in the main draft do not display the actual numbers, the tables in the Appendix reveal that the differences between models are, in fact, quite small. Therefore, it is strongly recommended to assess statistical significance. The conclusion that models know more when their size gets bigger is not very informative.
- The paper is somewhat difficult to follow and glosses over many important details, such as and similarity measurement with GPT-4 and prompts that were used in the experiments. Considering that similarity measure is a critical component of the discriminator-critique-gap score, it is crucial to provide comprehensive details and performance metrics in the draft. Additionally, there is room for improvement in the writing style.
- The framework's design raises concerns. Most importantly, it is unclear why we should regard GPT-4 responses to represent human values. There can be many directions for each values. Treating a single GPT-4 response as the ground truth for representing human values seems insufficient.

**Questions:**

- I fail to see the necessity of Section 2. It can be simply reduced to one sentence, which is the last sentence of Section 2: “If AI systems cannot understand the inherent intricacy and adaptability of values, their decisions may lead to adverse social consequences.”
- What is the purpose of Section 4.4? It appears to be redundant with the preceding content. VUM seems more akin to a dataset than a framework.
- There is a missing equation reference in the second sentence of Section 5.3.

---

### Official Review · Reviewer_3wE5 · 2023-11-04

**Soundness:** 2 fair
**Presentation:** 2 fair
**Contribution:** 2 fair
**Rating:** 3
**Confidence:** 4

**Summary:**

This paper explored the way to understand the values in LLMs. The authors argued that understanding the values in LLMs requires considering both "know what" and "know why". They presented the Value Understanding Measurement (VUM) framework that quantitatively assess both “know what” and “know why” by measuring the discriminator-critique gap related to human values. They collected data using the Schwartz Value Survey and use GPT-4 to measure the similarity between the models' response and humans'. The experimental results showed that (1) the ability of LLMs to understand values is greatly influenced by context rather than possessing this capability inherently; (2) The LLM’s understanding of potentially harmful values like “Power” is inadequate.

**Strengths:**

- This paper studies a very interesting problem: what values are hold in the LLMs. Understanding the correct way to probe and measure this will be beneficial to the whole community.
- To me, the way the authors proposed to achieve this goal is novel.

**Weaknesses:**

- I am confused by several parts when reading the paper:
  - In Figure 1, it's super unclear what is fed to the model in each step. Is the whole history included? It's more clear when reading through 4.3 but it would be much better if that can be reflected in the Figure. Also, it seems that the critique model only takes the two reasoning text as inputs. Then how does it deal with the contextualized phrases like "the values I mentioned"?
  - In 5.1.2 the authors mentioned they have a setting about "Add prompts that make it relevant to the values". I am also confused by this part. What are the prompts used here and where is the results? I would be better if the authors can "dry run" several examples for each setting in the appendix, which could help readers understand the whole process much easier.
  - I also had a hard timing parsing Figure 5.

- The authors claimed that "The scaling law (Kaplan et al., 2020) significantly impacts "know what" but not much on "know why", which has consistently maintained a high level". I am not convinced by the argument:
  - Model parameters are not the single factor that affect the models' performance. For example, how many tokens they have been trained on, and what alignment data/methods are used heavily affect models' performance. Furthermore, there is no official statement from OpenAI about the size of GPT-3.5-Turbo (rumor: 20B model) and GPT-4 (rumor: MOE of 200B model). How is that possible to make such a claim without knowing the details of the two models?
  - I am also not convinced by the "know why" argument. Seen from Figure 3 and Figure 4, all models achieve quite similar scores. However, could it be that the GPT-4 doesn't have the capability to distinguish those models' outputs from the baseline reason? I saw the authors mentioned they did some kind of human study in Appendix B.2, but all the details about this human study is missing. I would like to see a thorough human study and analysis about this part.

**Questions:**

see weakness

---

### Official Review · Reviewer_Lh4W · 2023-11-11

**Soundness:** 2 fair
**Presentation:** 2 fair
**Contribution:** 2 fair
**Rating:** 5
**Confidence:** 3

**Summary:**

This paper presents the Value Understanding Measurement (VUM) framework to quantitatively assess the understanding of values in large language models (LLMs) by measuring the discriminator-critique gap related to human values. The authors use the Schwartz Value Survey to specify evaluation values and develop a dialogue dataset with GPT-4. They evaluate five representative LLMs, including GPT-3.5-Turbo, GPT-4, LLaMA2-7B, LLaMA2-13B, Vicuna-33B, and find that the scaling law significantly impacts "know what" but not much on "know why". The results suggest that LLMs might craft plausible explanations based on context without truly understanding their inherent value, indicating potential risks. The paper contributes a comprehensive measurement system VUM, a dataset based on the Schwartz Value Survey, and new perspectives for value alignment in LLMs.

**Strengths:**

- The paper introduces a new framework, Value Understanding Measurement (VUM), for quantitatively assessing the understanding of values in large language models (LLMs). This is an innovative approach to evaluating the alignment of LLMs with human values.
- The authors provides a detailed methodology, including the use of the Schwartz Value Survey, the construction of a dataset, and the evaluation of five representative LLMs. This comprehensive investigation allows for a thorough analysis of the models' value understanding abilities and brings insights to the community.
- This paper offers valuable insights into the limitations of LLMs in understanding values, such as the influence of context, the scaling law's impact, and the potential risks associated with inadequate understanding of potentially harmful values, which might be interesting for the research community.

**Weaknesses:**

1. Evaluation dataset
- [major] The proposed dataset used for evaluation is generated by GPT-4. As a dataset for evaluation, the quality of this dataset is not confirmed. Also, it is uncertain whether this dataset could fully capture the complexity and adaptability of human values.
- [minor] Some details of the methodology are missing. For example, how do you construct the questions in the dataset? By GPT-4 or human annotators? I only find information on the answers  in ``there are baseline answers provided with GPT-4, along with explanations (baseline reasons) on why these answers correspond to their respective values.''

2. While the VUM framework is novel, it relies on the use of GPT-4 as an annotator for value similarity and reasoning ability. The reliability of this annotation method is not discussed. If there is evidence that the GPT-4 annotator has high alignment with human beings in the value similarity estimation and reasoning, then the conclusions will be more convincing.

**Questions:**

Please refer to the weaknesses.